# The Recent Floods in the Asso Torrent Basin (Apulia, Italy): An Investigation to Improve the Stormwater Management

**Marco Delle Rose** [1,*] 🆔**, Paolo Martano** [1] 🆔 **and Corrado Fidelibus** [2] 🆔

[1]   Istituto di Scienze dell'Atmosfera e del Clima, Consiglio Nazionale delle Ricerche, 73100 Lecce, Italy; p.martano@isac.cnr.it
[2]   Dipartimento di Ingegneria dell'Innovazione, Università del Salento, 73100 Lecce, Italy; corrado.fidelibus@unisalento.it
*   Correspondence: m.dellerose@le.isac.cnr.it

**Abstract:** Stormwater management is of concern to public institutions and academies. In the Asso Torrent endorheic basin (Salento peninsula, Southern Italy), a network of natural and artificial channels crossing urban and rural areas and flowing towards six swallow karst holes, several floods occurred in the last six years, after the end of extensive hydraulic works. In this paper, the results of an observational study on the meteorological and hydrological factors concurring to these floods are reported. It was inferred that soil saturation, cumulative precipitation anticipating the events, and clogging of the swallow holes are of relevance and must be considered in the definition of new criteria in decision-making procedure. The adoption of both innovative modeling techniques and real-time control should be an efficient solution to properly regulate the flow control devices before and during the precipitation events. With a view to providing solutions for a sustainable management of the water resources, easy-to-implement measures are suggested, such as the selection of flood-tolerant crops and construction of harvesting systems for alternative water uses.

**Keywords:** water resources management; floods; swallow holes; endorheic basin; real-time stormwater management; karst; Apulia (Southern Italy)

## 1. Introduction

Presently, because of climate change, increase of population and improper land use, stormwater management is of paramount importance in many regions of the world [1,2] and the relative solutions are of concern to public institutions and academies [3,4]. Specifically, semi-arid areas of the Mediterranean are particularly involved for the adverse effects of the current climate trend [5,6]. In this respect, in accordance with the EU Floods Directive 2007/60/EC, several EU state members, including Italy, have promulgated national plans to mitigate the risk of flood caused by stormwaters. With reference to the Apulia region (Southern Italy, central Mediterranean), previous studies highlighted the unpleasant effects of significant precipitation events at the basin scale [7–9].

For the Asso Torrent basin (Salento peninsula, Southern Apulia), stormwater management is a critical issue for the frequent occurrence of damaging floods, even though maintenance works on the channels of the basin have been accomplished for long [10]. Since the first works dated back to the 1920s, one of the main objectives was to prevent the floods. However, the results were not completely satisfactory, thus, essentially for the protection of the urban areas, in 2010–2013 more consistent works were executed that completely modified the lower course of the torrent. In addition, for the real-time stormwater management, sluice gates were set up to be operated for flow regulation. Since then,

the efficiency of the structures of the basin in preventing floods drastically changed [11]. It is worth remarking that for the stormwater management the resort to real-time control systems is generally increasing [12–15].

A side effect of the 2010–2013 works was the increase of the flood hazard in the sub-urban and rural areas; in fact, in the last six years, these areas were affected by four floods. Particularly detrimental to the agricultural activities was the flood in March 2015, which occurred after only moderate precipitation in the previous days. In addition, it was not predicted by the competent authority, therefore also questions on the efficacy of the alert system raised. It can be stated that the basin is presently affected by an hydraulic disorder. In this note, the results of an investigation about the meteorological and hydrological causes generating the hydraulic disorder in the Asso Torrent basin are reported with the aim to furnish a contribution to the solution of the problems concerning the stormwater management before and during precipitation events. The investigation approach used herein, essentially observational, is applied in order to define the relative importance of the factors concurring to the floods. The approach has well-established scientific bases in the research activity about flood hazards [16–18]. The paper also contains an examination of the site-specific conditions predisposing the basin to floods; it could be useful to prevent other hydraulic and environmental failures in the basin, as well as in other basins with similar regional meteorological and hydrological contexts.

The paper is organized as follows. In Section 2, the structure of the basin is described and a mention is given to four damaging floods occurred in the five-years period 2013–2018. In Section 3, the meteorological conditions in the Salento peninsula in that period are illustrated. In Section 4, the hydrometric data of the basin are reported. Finally, in Section 5, a discussion of the results of the investigation is provided for the benefit of the stakeholders of the stormwater management.

## 2. The Asso Torrent Basin

The Asso Torrent is a network of natural and artificial channels crossing urban and rural areas, flowing towards six swallow karst holes, locally named "vore" (singular *vora*, from Latin, means *swallow* or *whirlpool*). The karst holes are connected to a regional karst aquifer, affected by seawater intrusion, providing about 75% of the local water supply (the hydrogeological setting is described in [10,19]). The basin is endorheic (i.e., it retains runoff within the watershed) and extends over 220 km$^2$, excluding the area drained by an artificial sea channel (Figure 1). Maximum and average heights are 198 and 68 m, respectively, with an average slope of 2%. Several zones of the basin are clayey, including the lowest ones, thus it is particularly prone to floods despite the karst substratum.

The first hydraulic works were probably carried out in the medieval age; however, only in the 1920s, with the marshland reclamation for eradicating the malaria scourge, the current structure of the lowest area of the basin was set (Figure 2A,B). The first specific flood defense works began in the late 1970s and consisted of the construction of a new channel, connected to the sea and equipped with a double sluice gate (G1 in Figure 2B). Since 1991, the torrent has been used for the disposal of treated wastewater from five Sewage Treatment Plants (STPs). The disposal was conceived also to reduce the seawater intrusion in the karst aquifer [20].

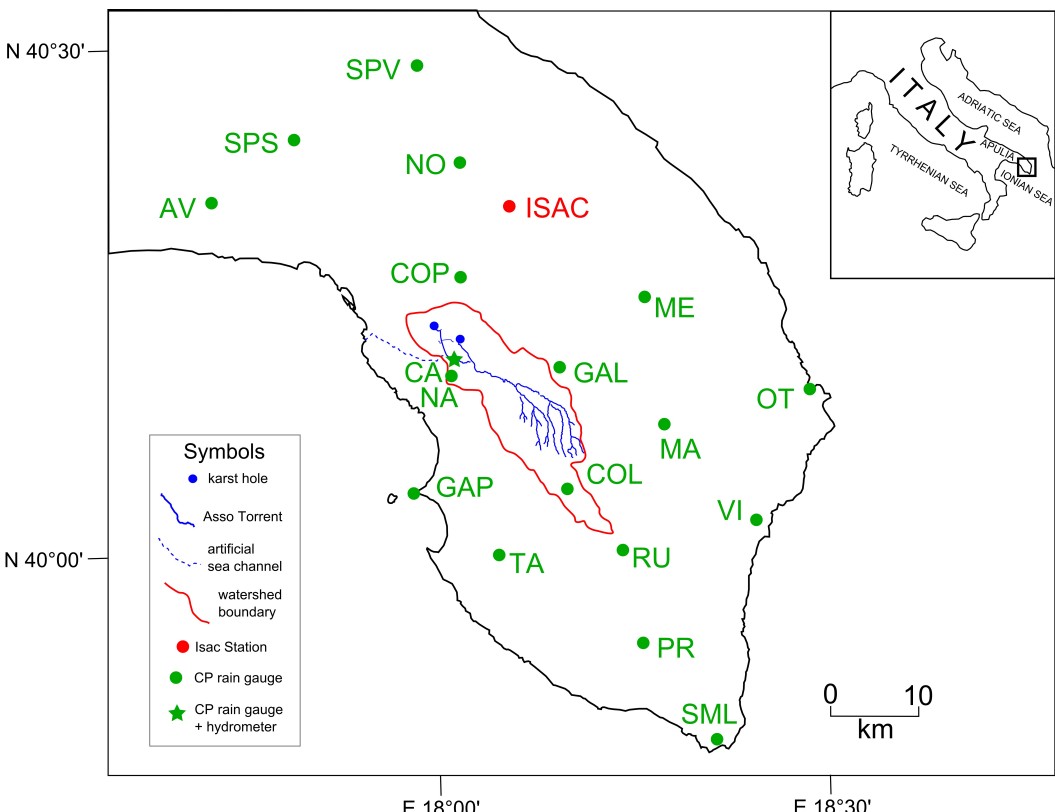

**Figure 1.** Map of Salento Peninsula and Asso Torrent basin; the measurement stations of the Civil Protection network are also shown (see the Abbreviations).

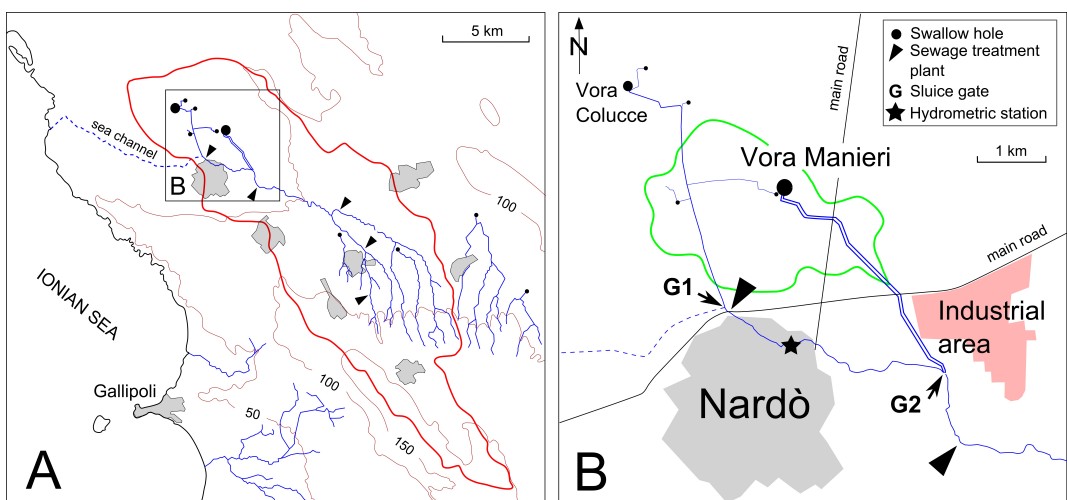

**Figure 2.** (**A**) Scheme of the Asso Torrent basin; (**B**) Detail of the lowest course of the torrent; the green line delimits the area affected by the 2013–2018 floods.

Until summer 2013, the main flow was discharged in the *Vora Colucce* swallow hole, a doline progressively clogged by the deposition of a brownish mud, containing a significant amount of organic matter because of the insufficient dimensions of the STPs, being their total capacity corresponding to only 72.5% of the served population; as a consequence, the discharge capacity of the swallow hole reduced from 10 m$^3$/s to 1 m$^3$/s in thirty years [11]. In the meantime, storms of increasing severity occurred with peak flows in the basin of about 30 m$^3$/s, thus runoff problems were experienced. Especially in the urban area of Nardò (Figure 2), one of the largest towns in Salento peninsula, severe damages were recorded such as sewer system spills, destruction of commercial goods, damage to

local agricultural activities and flooding of basements and first floors. It must be noted that *Vora Colucce* would be insufficient to drain stormwater runoff even if there were no clogging, given that the maximum return time of the rainfall event not producing a flood with the initial discharge capacity is 10 years only [11].

As previously mentioned, in 2010–2013 period, new hydraulic works modified the lower course of the torrent. For the flood prevention in the urban areas, a new channel was built diverting the flow by means of a sluice gate (G2 in Figure 2B). Unlike the cemented channel of *Vora Colucce* swallow hole, the new channel is permeable, thus the chance of flood nearby is reduced. It was designed by considering a maximum discharge of almost 200 m$^3$/s for a return period of 200 years. Due to a spillway, the new channel acts as an overflow channel (i.e., it becomes active when the upstream G2-hydrometric level exceeds a given threshold) and is connected to a minor swallow hole, the *Vora Manieri* hole, with a discharge capacity of 2 m$^3$/s (Figure 3).

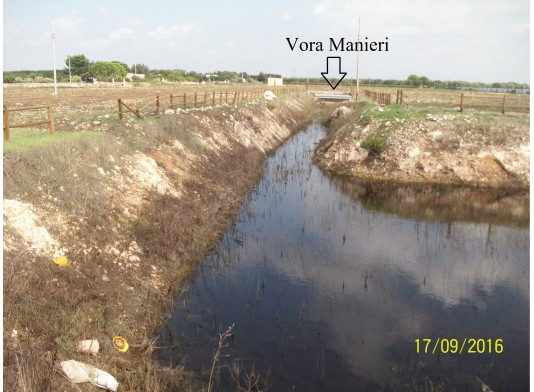 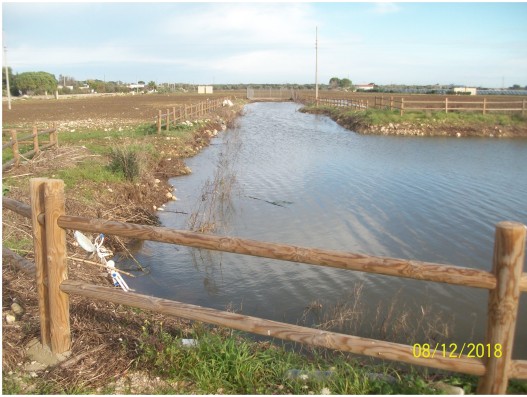

**Figure 3.** Hydrometric level of the final course of the new channel of the Asso Torrent; left: low after the first rains of the beginning of the 2016 wet season; right: high at the onset of a flood, resulting from moderate precipitation before 8 December 2018.

The 2010–2013 works transformed the Asso Torrent into a sort of centralized open-channel infrastructure; the control of devices G1 and G2 is now a tool for the decision-making during storms. However, the stormwater management at the basin is still troublesome, as demonstrated by the recent floods affecting the sub-urban and rural areas surrounding the *Vora Manieri* hole. Four floods (F1 to F4) were particularly significant and are analyzed in the note.

In Table 1, the dates of these floods are reported, together with the estimated degrees of the damage and the duration of the triggering precipitation. The average duration of the floods was one week.

**Table 1.** Floods F1–F4, relative degrees of damage in the lowest course of the torrent and main precipitation events anticipating or not anticipating the floods.

| Flood ID and Date | Degree of Damage | Precipitation Events |
|---|---|---|
| F1—1 December 2013 | low | 12/11/13–02/12/13 |
| no flood | | 23/01/15–27/01/15 |
| F2—7 March 2015 | middle | 22/02/15–06/03/15 |
| no flood | | 01/09/16–14/09/16 |
| no flood | | 04/01/17–18/01/17 |
| F3— 27 February 2018 | low | 11/02/18–28/02/18 |
| F4—23 October 2018 | low | 01/10/18–23/10/18 |

Floods F1, F3 and F4 caused limitations on the service of vehicles (low damage degree). F2 instead caused damages also to houses and agricultural activities and products, because it occurred at the beginning of the sowing period and lasted almost eleven days, thus reducing the early root development (Figure 4). F2 and F3 were not predicted by the competent authority.

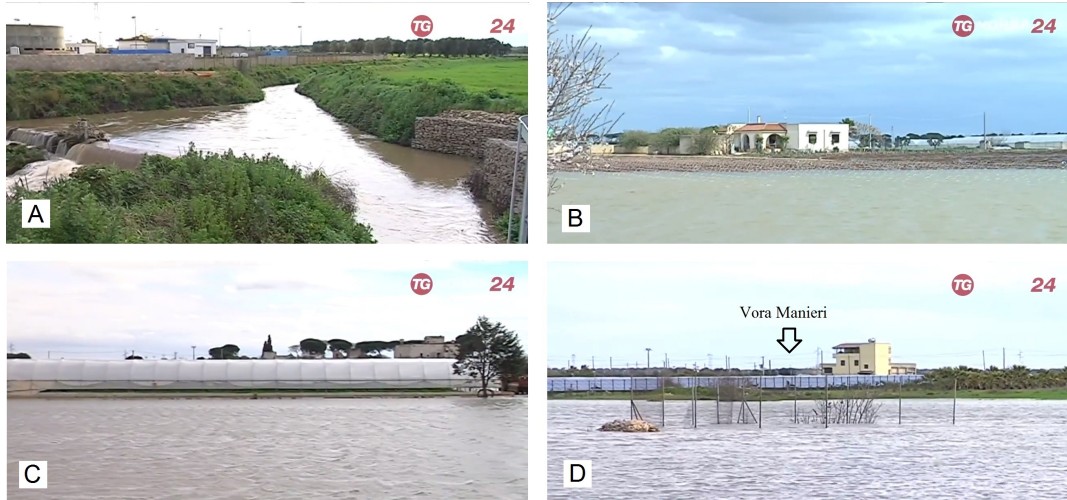

**Figure 4.** Pictures of the flood on 7 March 2015 [21]; (**A**) the torrent at the G2 sluice gate; note the overflow at the spillway on the left; (**B**) the flood reached a sub-urban zone; (**C**) flood damages in an agricultural factory; (**D**) the area around the swallow hole.

The magnitude of F2 was such to threaten the safety of the industrial area, located at the right side the new channel (see Figure 2B). The growing concern about the flood risk was motivated on 16 February 2018, when, as a consequence of a local flood in the middle course of the torrent, two people trapped in a car were rescued by a Civil Protection team. This event was emblematic of the persistent hydraulic disorder still affecting the entire basin, despite the many engineering works carried out.

No hydrometric station is located along the new channel; the only available data come from the instrument placed at the *Canale Asso* station (CA in Figure 1), located along the channel linked to *Vora Colucce* hole (Figure 2B). This station was built in 2009 and was used until 2012 in the context of the urban stormwater management. During the huge flood of 2 November 2010 [10], a peak of 1.7 m was measured. After the construction of the new channel, the *Canale Asso* hydrometric station seems to have lost significance for the urban flood defense. Besides the direct economic and social damage due to the floods, a further consequence of the persisting hydraulic disorder affecting the Asso Torrent basin (especially around *Vora Manieri* hole) regards the deterioration of the underground water quality and the harm to the public health. In fact, given that the STPs are undersized, the water flowing out of the channels may contain dangerous pollutants not filtrated by the karst vadose zone on top of the aquifer [10]. Also for this reason is it therefore necessary to find new effective and sustainable solutions to improve the stormwater management of the basin.

## 3. Meteorological Features

In this section, the precipitation events related to the four floods are described, together with the other major precipitation events in the period 2013–2018 (Table 1). The data are extracted from the database of the micrometeorological station of ISAC-CNR (Istituto di Scienze dell'Atmosfera e del Clima, Consiglio Nazionale delle Ricerche) [22]. The station is approximately located in the center of the peninsula (ISAC in Figure 1) and daily collects a complete set of 30-min averaged micrometeorological and soil variables [23]. The areal average precipitation in the selected period have been calculated from the records of the rain gauges (see Figure 1 for the locations), belonging to the regional Civil Protection network. The archive maps of the model cascade GLOBO-BOLAM-MOLOCH of the ISAC-CNR have been used to provide a meteorological overview of the selected events [24].

### 3.1. F1, 11/11/2013–2/12/2013

This precipitation event was characterized by the passage of a series of troughs coming from Northern Europe on the Mediterranean area, as shown in the 500 hPa GeoPotential Height (GPH)

maps of the BOLAM model. Several cut-off lows were then fed by the western moving troughs, thus generating a long instability sequence in the Mediterranean sea, lasting the whole second half of November 2013, and generally associated with intense surface lows (Figure 5).

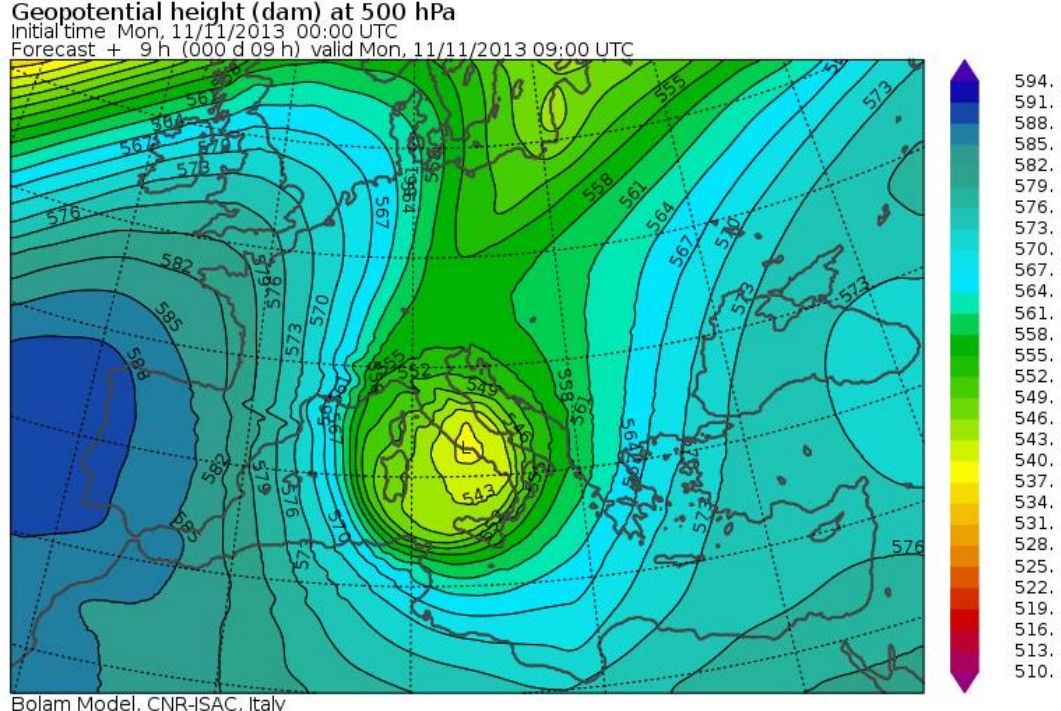

**Figure 5.** The starting conditions of the 500 hPa GPH of the precipitation events associated with flood F1 (see Table 1).

The last days of November the vorticity field intensified over the Atlas mountains and was followed by the generation of a short-lived Medicane-like surface cyclonic structure [25], impinging on the coast of Sicily on December 1 and causing enhanced precipitation on Southern Italy. The ISAC-CNR station recorded 270 mm from 11 November to 2 December and 200 mm from 19 November to 2 December with 116 mm in the first day. The MOLOCH model [26] forecasted about 250 mm from 12 November to 2 December, and about 200 from 19 November to 2 December. The average value of the 19 selected stations (including ISAC) was 247 mm with a spatial standard deviation of 32 mm and a ratio $r$ between standard deviation and mean equal to 0.13 that can be considered a proxy for the spatial variability of the whole precipitation event. From 17 November to 2 December, the areal averaged precipitation was 164 mm with a standard deviation of 25 mm and $r = 0.14$. In Table 2, the measured total precipitation for the flooding events are shown for the selected surface meteorological stations.

**Table 2.** Total precipitation (mm) measured at the ISAC-CNR station and at the rain gauges of Figure 1, during the four floods herein considered; see the Abbreviations.

|  | AV | SPS | SPV | NO | COP | ME | NA | CA | GAP | GAL | MA | OT | COL | TA | RU | VI | PR | SML | ISAC |
|---|---|---|---|---|---|---|---|---|---|---|---|---|---|---|---|---|---|---|---|
| **Flood 1** | 206 | 210 | 245 | 237 | 259 | 230 | 248 | 251 | 227 | 247 | 235 | 300 | 237 | 204 | 293 | 242 | 328 | 236 | 270 |
| **Flood 2** | 126 | 142 | 133 | 146 | 141 | 142 | 161 | 161 | 158 | 188 | 177 | 212 | 193 | 14 | 204 | 201 | 159 | 133 | 164 |
| **Flood 3** | 117 | 113 | 139 | 155 | 155 | 186 | 174 | 163 | 169 | 206 | 237 | 271 | 208 | 184 | 264 | 255 | 245 | 268 | 180 |
| **Flood 4** | 78 | 88 | 82 | 85 | 122 | 134 | 164 | 172 | 137 | 307 | 141 | 169 | 165 | 156 | 228 | 172 | 157 | 250 | 118 |

The climatic annual average precipitation in Southern Salento is about 650 mm [27], so that the total for the selected period represents more than 35% of the average annual precipitation in less than three weeks. The soil moisture at 30-cm depth recorded at the ISAC-CNR station (reported here as

a proxy for the bulk soil saturation conditions) was more than 0.30 m$^3$/m$^3$ (relative volumetric soil moisture $R$ about 70%, see definition later) before the beginning of the precipitation. Floods were recorded both in the lower and in the middle course of the Asso Torrent.

### 3.2. F2, 22/2/2015–6/3/2015

During the two weeks just before the flood, the weather conditions over Italy and the southern regions in particular were characterized by the passage of a series of low pressure vortices at sea level associated with a sequence of troughs of GPH at 500 hPa, a meteorological condition typically associated with relevant precipitation events. The sequential trough events peaked on 22 February, 25 February and 5 March, but the first two events actually showed continuous trough conditions extended from 22 February to 28 February, with prolonged rainy conditions in the region. The total precipitation on the area forecasted by the MOLOCH model for this sequence of events was about 200 mm. Like the 2013 event, the precipitation during the selected period looks relatively homogeneous for the selected stations, with an average value of 162 mm, a standard deviation of 26 mm and $r = 0.16$ for the whole period. The ISAC-CNR station recorded 164 mm as total precipitation for this event (Table 2). The total precipitation in about two weeks was more than 20% of the average annual precipitation. In Table 2 it can be noticed that most of the strongest precipitation of the event preceding the flood are located in the middle and upper course of the Asso Torrent (roughly corresponding to the area where the station MA = Maglie; COL = Collepasso; and GAL = Galatina are placed, see Figure 1). The soil moisture at 35-cm depth recorded at the ISAC-CNR station was 0.35 m$^3$/m$^3$ ($R$ about 75%), just before the beginning of the precipitation.

### 3.3. F3, 11/2/2018–28/2/2018

For this event, a total of 180 mm was recorded at the ISAC station (Table 1) with an average of 194 mm, a standard deviation of 50 mm and $r = 0.26$ in all the stations. The MOLOCH model forecasted about 250 mm on the Salento peninsula with an instability caused by a sequence of two troughs sequentially moving eastward and southward. They caused more persisting but not exceptionally intense precipitation over the region, accumulated in the periods 10–15 February and 20–27 February, respectively. The soil moisture at 35-cm depth recorded at the ISAC station was above 0.40 m$^3$/m$^3$ (90% of the soil moisture corresponding to full saturation) before the beginning of the precipitation.

### 3.4. F4. 1/10/2018–23/10/2018

The ISAC-CNR station recorded 118 mm for this period. The selected 19 surface stations (SML, Santa Maria di Leuca rain gauge, was not working in this period and data were obtained by the closest available station) showed an average of 153 mm, with a standard deviation of 59 mm and $r = 0.39$. About 160 mm over the peninsula were forecasted by MOLOCH. As shown by the GLOBO model, a trough from Northern Europe generated a cut-off of increasing force over the Genoa Gulf, and translating towards Southern Italy in the first days of October. On 20 October, a new trough from North East Europe generated a new cut-off North East of Italy moving westward and southward (see Section 5). The Scandinavian cold air at 500 GPH contrasting with the still warm Mediterranean surface was able to generate strong convective events. Actually, the precipitation was apparently less uniform over the area than the other events, characterized by a larger value of $r$, and also in time, diffused in a period longer than 20 days with a sequence of several dry days in between. Several stations recorded a total of less than 100 mm; however, a strong peak was recorded in Galatina (middle course of Asso Torrent). In addition, the soil moisture at 35-cm depth recorded at the ISAC station was 0.12 m$^3$/m$^3$ before the beginning of the precipitation (October 1), much less than in the previous floods ($R$ 25% only).Such a low value may be expected at the beginning of October in the area. Also on 21 October, before the last strong precipitation, the soil moisture was low ($R$ about 45% only). For this last event only, an areal average of 97 mm with a standard deviation of 51 mm and $r = 0.52$ was recorded, mainly between October 22 and October 23. The ISAC-CNR stations collected 59 mm, and a peak of 245 mm in

two days was recorded in Galatina (GAL in Figure 1). These values are typical of a strongly convective event, bringing a huge localized precipitation in a short time. This precipitation event was associated with the second flood of the Asso Torrent basin in 2018.

*3.5. Other Events*

The other relevant events (not associated with a flood but quantitatively comparable to the ones that triggered F1–F4), registered by the ISAC-CNR station in the last six years, are the following:

- 23/1/2015-27/1/2015—The ISAC-CNR station recorded a total of 79 mm and about 100 mm were forecasted by MOLOCH. The event was caused by a deep trough hovering above the Mediterranean area giving rise to a cut-off in Southern Italy. The initial soil moisture at 35-cm depth recorded at the ISAC station was over 0.40 m$^3$/m$^3$ (*R* about 90%).
- 1/9/2016-14/9/2016—A total of 180 mm was recorded at the ISAC-CNR station and about 185 mm were forecasted by the BOLAM model over Salento Peninsula. The event was associated with a sequence of two cut-off lows at 500 hPa. The second strongest low-pressure center remained for more than one week over Southern Italy, and  mainly triggered strong convective precipitation in the region. However, the soil moisture at 35-cm depth recorded at the ISAC station was less than 0.20 m$^3$/m$^3$ (*R* about 40%) before the beginning of the precipitation.
- 4/1/2017-18/1/2017—A total of 148 mm was recorded by the ISAC-CNR station and about 200 mm were forecasted by MOLOCH. In this case a deep trough extended to Southern Europe generated a sequence of three deep low-pressure centers in the Mediterranean Sea that persisted on the Italian Peninsula for many days. The initial soil moisture at 35-cm depth recorded at the ISAC station was above 0.30 m$^3$/m$^3$ (*R* about 70%).

## 4. Hydrometric Data

As aforementioned, the only hydrometric station located within the Asso Torrent basin is placed along the channel linked to the *Vora Colucce* hole (Figure 2B). Since autumn 2013, only a small amount of flow is diverted in this channel because of a spillway (located just upstream the gate G2, see Figures 2B and 4A) that was designed to convey the largest part of the flood volume in the new channel. Unfortunately, measurements were not collected before 2016. In what a follows, the data of the *Canale Asso* hydrometric station from 2016–2018 are analyzed to look for possible correlations with: (a) the second and the third precipitation events not causing flood; and (b) floods F3 and F4 (Table 1). In these three years, the water level was around 0.2 m during the dry days. This level is maintained during the entire summer season for the discharge of the treated water of the STPs.

A first peak exceeding 0.3 m during the 1/9/2016–14/9/2016 event was recorded on 6 November; on 9 September the maximum of 0.96 m was reached and, finally, on September 13 the last peak of 0.82 m was measured (Figure 6). Both these peaks were reached in about two hours. Unlike this case, the 4/1/2017–18/1/2017 precipitation event showed no sharp peaks. However, the level kept higher than 0.2 m for many days and on the last day it reached 0.45 m.

As far as F3 is concerned, on 16 February 2018, the hydrometric level reached 0.5 m and then returned to 0.2 m within one day. In the same time, a huge local flood (see Section 2) occurred at the middle course of the torrent. At the end of 25 February, the water level rose to 0.45 m (within six hours) and then remained higher than 0.40 m until the early hours of February 27, the day of the flood. Finally, on March 1 the level returned to the average value (Figure 6).

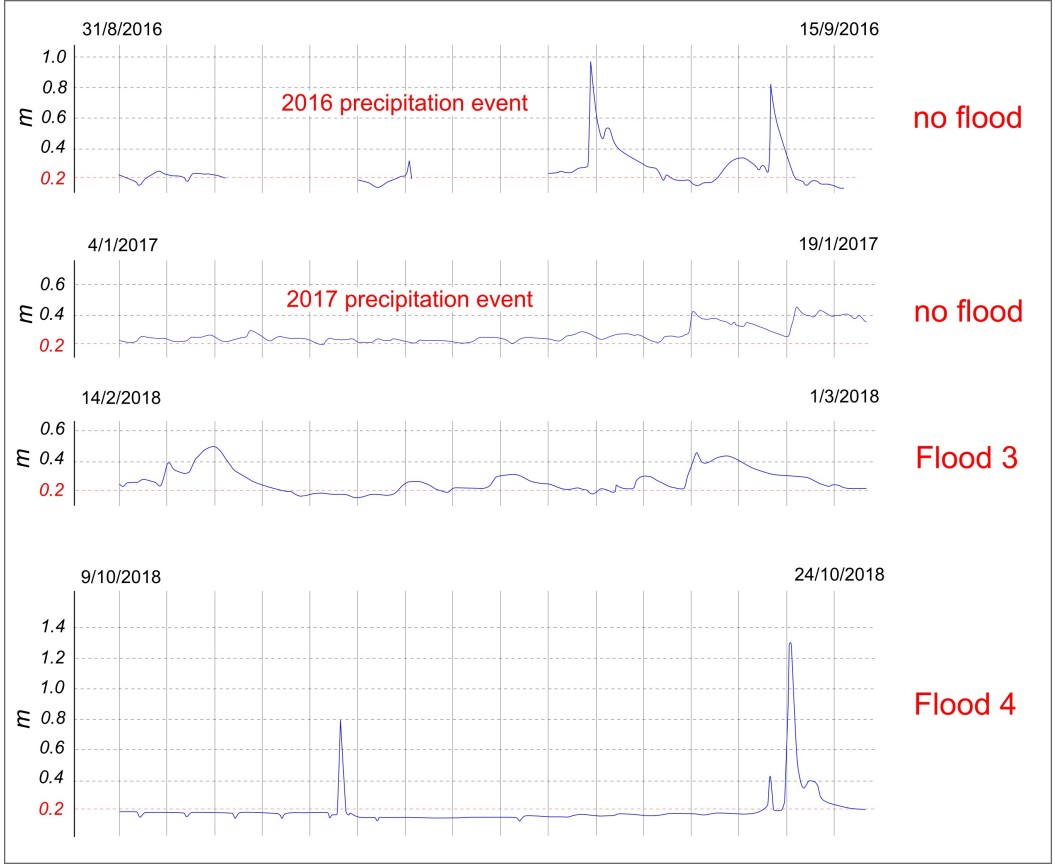

**Figure 6.** Hydrometric levels (m) recorded at the *Canale Asso* station (the rough data have been extracted weekly from [28]). Precipitation events and floods are reported in chronological order from top to bottom (see Table 1).

During the weeks preceding F4, the trend of the water level was quite different compared to the previous one. After several days with levels around 0.2 m, a sharp peak of 0.8 m occurred on 13 October; rapidly, the water then returned to 0.2 m and remained at this value until the first half of the day previous the flood. A peak of about 0.4 m was recorded in the afternoon of 22 October, while at the same time as the flood, the hydrometric level quickly (two hours) reached 1.3 m, then remained higher than 0.4 m for fourteen hours. It has to be noticed that the floods of 2018 were associated with both short peak events, with dry soil, and prolonged high level conditions with saturated soil (Figure 6).

## 5. Discussion

The investigated floods were mostly related to meteorological troughs in the GPH at 500 hPa prolonging southwards from the Northern Europe, and insisting over the Mediterranean basin for several days, due to relatively high pressure conditions placed eastward and westward of the same troughs. In particular, F4 (see Table 1) was related to a Scandinavian trough generating a cut-off low, anomalously moving eastward and southward through the Balkans (Figure 7). These meteorological events are generally associated with high pressure blocking in the Northern Atlantic (negative phase of the North Atlantic Oscillation, NAO), often acting together with a Siberian high pressure center [29]. It has been pointed out that climatic configurations with two positive anomalies of the 500 hPa GPH, located over the North Atlantic and the Siberian Plateau, became increasingly frequent in recent decades as a signature of the climate change [30]. In the short/mean term, this circumstance and the observation of a possible increasing trend of quick strong precipitation events over the Salento peninsula, especially in the dry season [19], suggest diminishing return times for the floods in the basin.

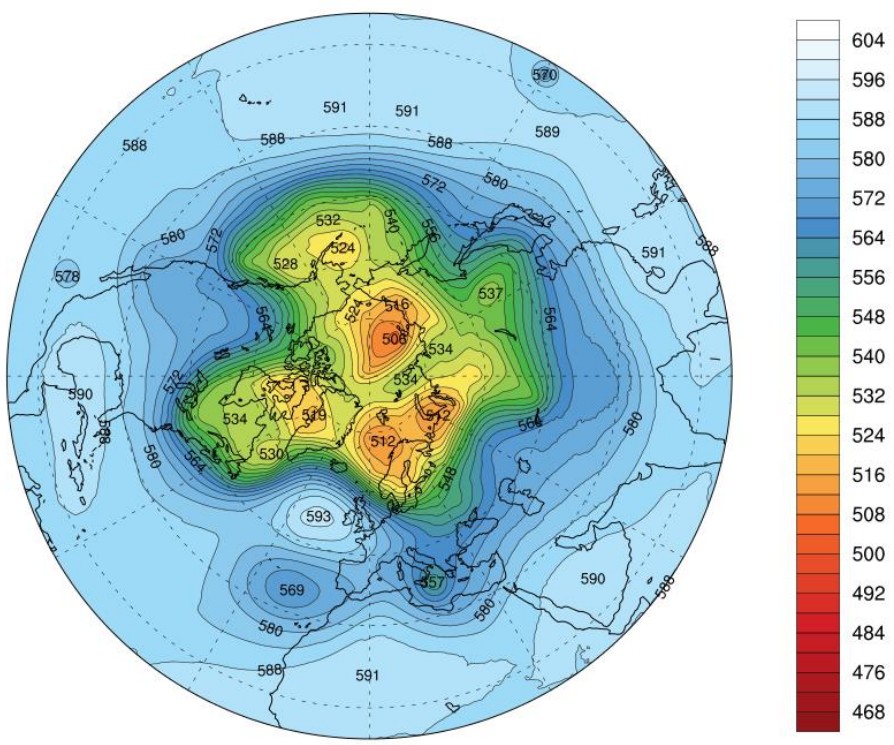

**Figure 7.** Northern hemisphere map of the 500 hPa GPH associated with the last strong precipitation of the 2018 second flood (i.e., the ID F4 of Table 1).

It is widely recognized that the initial soil moisture is an important hydrological property to be evaluated when dealing with the risk of flood [31]. The chance of flood increases when the soil is fully saturated or close. Other factors may trigger or concur to a flood; however, in what follows the association of precipitation with initial soil moisture is considered for the interpretation of the flood episodes. Soil moisture is quantitatively defined herein as relative volumetric soil moisture $R$, i.e., the ratio of the volumetric soil moisture $\theta$ with the corresponding value for full saturation $\theta_s$, this quantity equal to the soil porosity.

In Table 3, for each main precipitation event in the 2013–2018 period, the total cumulative precipitation $P$ (averaged from all the local available stations of Figure 1 for the flood episodes or estimated by both ISAC station and Moloch model for the other cases, in mm), the initial relative volumetric soil moisture $R_0(\%)$, the time-averaged (over the whole event) value of $R(\%)$, say $\overline{R}$, the number $N$ of times $\theta$ resulted equal to $\theta_s$ ($R=1$) during the whole event and the occurrence of flood are reported.

**Table 3.** Meteorological/hydrological values measured during the main precipitation events in the 2013–2018 period; $P$(mm) total cumulative precipitation (spatial average and standard deviation or estimated interval, see text), $R_0$(%) initial relative volumetric soil moisture, $\overline{R}$(%) time-averaged (over the whole event) value of the relative volumetric soil moisture $R$, $N$ number of times the volumetric soil moisture reached the value corresponding to full saturation during the event, occurrence of flood.

| Date | $P$ (mm) | $R_0$ (%) | $\overline{R}$ (%) | $N$ | |
|---|---|---|---|---|---|
| 12/11/13-02/12/13 (F1) | $247 \pm 32$ | 70 | 80 | 1 | yes |
| 23/01/15-27/01/15 | $80 \div 100$ | 90 | 90 | 0 | no |
| 22/02/15-06/03/15 (F2) | $162 \pm 26$ | 75 | 90 | 2 | yes |
| 01/09/16-14/09/16 | $180 \div 185$ | 40 | 70 | 2 | no |
| 04/01/17-18/01/17 | $150 \div 200$ | 70 | 80 | 0 | no |
| 11/02/18-28/02/18 (F3) | $194 \pm 50$ | 90 | 90 | 1 | yes |
| 01/10/18-23/10/18 (F4) | $153 \pm 59$ | 25 | 60 | 1 | yes |

The data concerning soil moisture are limited (one station only) and refer to a site outside the basin, thus any conclusion has to be advanced with caution; however, with the exception of F4, the flood episodes were characterized by high values of $P$ (>150 mm), $R_0$ (>75%) or $\overline{R}$ (>75%) and the full saturation was reached at least one time during the precipitation. The no-flood events lack at least one of these features. A discrepancy with respect to F1–F3 was experienced for F4, where $P$ assumed a value even less than in some no-flood events and the values of both $R_0$ and $\overline{R}$ were far from full saturation; however full saturation was reached one time. The strong precipitation peak localized in Galatina (about 300 mm totally with 250 mm concentrated between October 22 and October 23) might have triggered this flood; however, the soil was almost dry and the event was the last in the 2013–2018 period, thus a decrease of the initial discharge capacity of *Vora Manieri* hole can be advanced as hypothesis (2 m$^3$/s, see Section 2). It is corroborated by the occurrence of brownish mud at the final course of the new channel (Figure 2B), detected in recent field surveys (Figure 8), and supposedly similar to the mud causing the clogging of the *Vora Colucce* hole [11]. Future speleological inspections, mud sampling and physical-chemical analyses will help in validating the hypothesis; however some justifications can be provided in what follows.

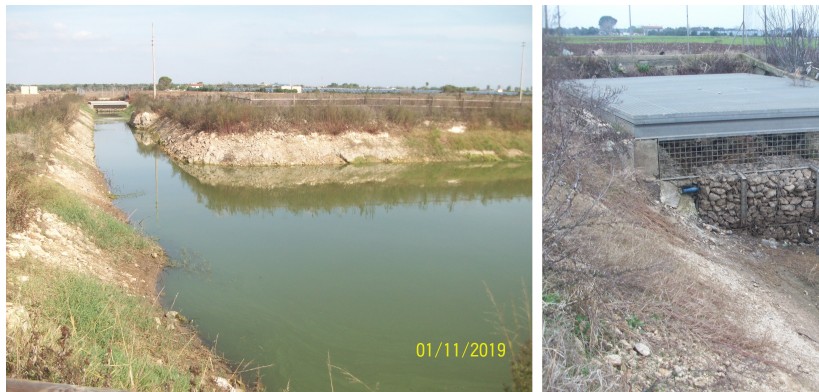

**Figure 8.** Left: the final course of the new channel in the beginning of November 2019; the hydrometric level was at a mean height, despite only 115 mm of rain in the previous four months; right: the entrance of *Vora Manieri* hole in the middle of January 2020; the brownish mud that covers the channel bed could cause clogging processes in the swallow hole.

The reduction of the discharge capacity of the *Vora Colucce* swallow hole begun in the early 1990s just after the disposal of treated wastewater in the torrent (Section 2). In particular, the measured discharge capacity reduced from 10 to 1 m$^3$/s resulting from a gradual process of clogging affecting the karst fracture system around the hole [11]. The clogging is produced by the interaction among chemical, physical, mechanical, and biological processes [32]. In karst systems, because of the adhesion

of organic and inorganic particles to fractured walls, such a reduction of capacity may be associated with the clogging of a limited number of fractures or karst conduits [33,34]. After 2013, no further works were accomplished in order to adapt the STPs to the treatment of the increased volumes of wastewater (Section 2), therefore a similar reduction of capacity of the new *Vora Manieri* receptor for these clogging processes is highly probable.

The dataset of the main floods affecting the lower course of the Asso Torrent before 2013 was described in previous works [10,11]. Even in consideration of the modifications of the course for the last hydraulic works, the past flood events can be compared to the ones described herein. As documented in newspaper and technical reports since 1947, no event was characterized by precipitation similar to those of Table 1. The floods that took place in the first 25 years from 1947 were associated with precipitation with *P* values less than 100 mm. Later on, a drastic change is evident marked by irregular cumulative rains and peaks of rain the day before the flood up to 180 mm (Figure 9).

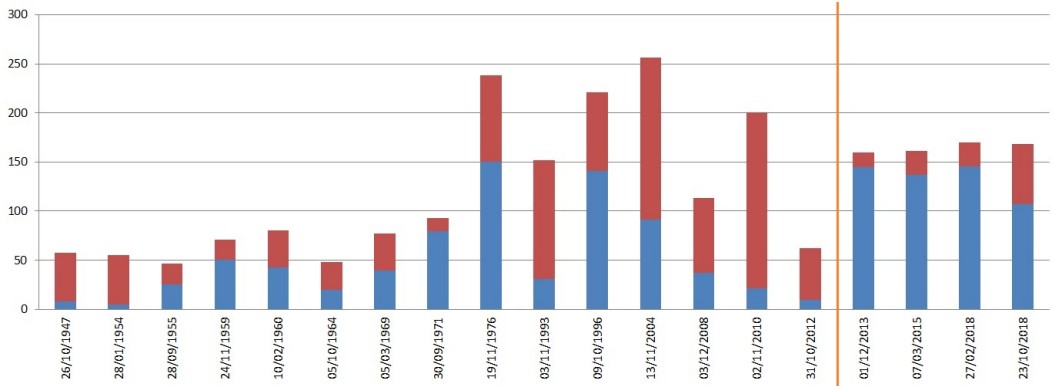

**Figure 9.** Two-weeks cumulative precipitation (mm) anticipating floods in the Asso Torrent basin (Nardò rain gauge); red segments refer to the rain the day before; data from *Annali Idrologici* of the Italian "Ufficio Idrografico e Mareografico" [35]. The orange line marks the end of the last extensive hydraulic works.

It must be noticed that the excavation of the sea channel activated by the sluice gate G1 (Figure 2) was executed in the 1970s, in conjunction with a change in the rain pattern causing the floods. Cumulative precipitation around 150 mm during the two weeks before the floods (i.e., an amount similar to the ones of the 2013–2018 events) were recorded only for the episodes of 19 November 1976 and 9 October 1996, respectively. As far as the floods of 3 November 2008 and 31 October 2012, are concerned, the severe damages reported in both the episodes would be consequent to wrong maneuvers of the flow control devices [10], as confirmed by the low cumulative precipitation values before the floods (Figure 9).

In view of the solution of the current problems of stormwater management, one may state that the reduction of the discharge capacity of the swallow holes in conjunction with the probable change of the precipitation pattern would require, first and foremost, an upgrade of the decision-making process that should rely also on the measurements of cumulative precipitation (for a preselected period) and soil moisture, rather than on the measurement of rain intensity only. Currently, the hydrological management of the Asso Torrent consists of the opening/closing of the sluice gates G1 and G2 (Figure 2B) and, in relation to specific meteorological conditions, aims at fulfilling the following requirements: prevention of urban floods, containment of seawater intrusion in the karst aquifer, groundwater pollution reduction [10,11]. In consideration of this investigation, special care must be taken when using the sluice gates if the soil moisture is high, even if low precipitation are forecasted.

However, with the increasing clogging of the swallow holes, less and less cumulative precipitation would be sufficient to cause floods. In this respect, the adjustment of the STPs in consideration of the actual served population constitutes the main countermeasure. Moreover, it must be also stressed that a correct functioning of the STPs would reduce also the risk of groundwater pollution. The Salento

peninsula is a warm temperate area with drought-prone summers (Csa climate type described by Peel et al. [36], i.e., temperate dry-hot summers) and, as other areas of Southern Italy, at risk of desertification [37]. Thus, the only drinkable water resource of the Salento peninsula, the deep karst aquifer, is subjected to an increasing water stress, especially as a consequence of the growing demand during the dry season. The area affected by the floods herein considered borders with a pollution-protected zone where the deep aquifer is consistently exploited. Again, in the surrounding of the *Vora Manieri* hole, where the groundwater is extensively pumped for irrigation, harm to human health may derive [10,20]. A runoff regulation reconciling the safety of public health with the need for recharging of the aquifer is still paradoxically required for the water management of the Asso Torrent basin, after one century from the marshland reclamation works. However, in consideration of the incoming challenges, advanced treatment processes prior to injecting treated water into karst aquifers are mandatory, especially in case of human consumption [38,39].

It must be pointed out that the assessment of a rainfall threshold is generally an hard task. Several methods are used for exorheic basins, based on the comparison of the relative performance [40], or on the measurements of the soil moisture [41]. In karst endhoreic basins, such as the Asso Torrent basin, there is no outflow to external water bodies and the outlet of water course is strongly controlled by the discharge capacity of the swallow holes, therefore, if clogging processes take place, the assessment of a threshold is uncertain.

Currently, a real-time control seems the only effective method to support the stormwater management of the Asso Torrent basin. If urban floods do not occur, the containment of seawater intrusion (diverting the flow toward the *Vora Colucce* hole) or the reduction of groundwater pollution (diverting the flow toward the sea) may be fulfilled [10]. The first objective is crucial during the dry periods with intensive pumping, while the second objective must be pursued when the efficiency of the STPs is low. Under these conditions, the Asso Torrent would be subjected to an innovative stormwater management, addressed not only to ensure the safety of citizens and protect the material goods during rainstorms but also to contribute to the improvement of quantity and quality of the water resources [14,42,43]. For the regulation of the gates before and during floods, the management stakeholders must establish optimal real-time operations, because the long-term approach cannot be used, due to the short timeframe of the new course of the torrent. Some modeling techniques used in a real-time control can be adopted to adjust the operations when facing single precipitation events [44].

In a medium-long term perspective, as pointed out in recent contributions (see e.g., [45]), also for the Asso Torrent basin, the need for a holistic transdisciplinary viewpoint for a sustainable water management emerges. Measures should be taken to convert agricultural practices in the affected area in favor of flood-tolerant crops. Moreover, the adoption of distributed treatments in urban areas, such as the greywater recovery, may provide a steady source of service water during the dry season. Nevertheless, other measures must be taken to counteract the hydraulic disorder of the basin. By considering the hydro-geomorphological features, the construction of decentralized stormwater infrastructures (especially rainwater harvesting systems) within the upper and middle course of the torrent may constitute an efficient policy action in order to reduce the runoff around the swallow holes. Again, the engineering updating of buildings and structures toward sustainable practices (such as the restoration of old rooftop harvesting abandoned tens of years ago in favor of centralized systems) is suggested. The construction of large harvesting systems for non-potable applications, designed for example for waste treatment facilities, can be also considered [46]. The resort to harvesting plants, besides the reduction of the hydraulic disorder within the basin, would also allow the containment of the demand of water supply, thus reducing the exploitation of the deep aquifer and, consequently, the seawater intrusion. Finally, the building of *engineered sinkholes* [47], upstream the points of discharge of the STPs, conceived to both reduce the basin runoff and increase the aquifer recharge, could represent another efficient policy action.

**Author Contributions:** Conceptualization, methodology, investigation, M.D.R., P.M. and C.F.; data curation of the ISAC micrometeorological station, P.M.; regional rain data, P.M. and M.D.R.; hydrological analysis, M.D.R. and C.F.; writing and editing, M.D.R., P.M. and C.F. All authors have read and agreed to the published version of the manuscript.

**Funding:** This research received no external funding.

**Acknowledgments:** This work is a contribution to the international HyMeX Program, with the partial support of the Italian PON I-AMICA Project.

**Conflicts of Interest:** The authors declare no conflict of interest.

## Abbreviations

Abbreviations and corresponding locations of Figure 1 and Table 2 are:

AV = Avetrana; CA = Canale Asso; COL = Collepasso; COP = Copertino; NA = Nardò; NO = Novoli; GAL = Galatina; GAP = Gallipoli; MA = Maglie; ME = Melendugno; OT = Otranto; PR = Presicce; RU = Ruffano; SML = Santa Maria di L.; SPS = San Pancrazio S.; SPV = San Pietro V.; TA = Taurisano; VI = Vignacastrisi.

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
