# Peer review of "The Recent Floods in the Asso Torrent Basin (Apulia, Italy): An Investigation to Improve the Stormwater Management"

_water, doi:10.3390/w12030661_

Round 1

Reviewer 1 Report

The manuscript examines the results of an investigation about the meteorological and hydrogeological causes generating the hydraulic disorder in the Asso Torrent basin, Southern Italy. It is worth to note that the Asso Torrent Basin is a karst region. Consequently, the manuscript examines a topic of interest to several regions worldwide with similar characteristics.

In my opinion, the authors have made an important and serious effort to improve the quality of the manuscript in relation to the previous version (manuscript ID:water-684474). I consider they succeeded it.

However, the manuscript is needed minor revision to be suitable for publication. Consequently, I recommend acceptance of the manuscript after minor revision.

Specifically:

The manuscript has to be written according to the instructions for Authors of the WATER Journal, namely in the reference list the Journal’s name in abbreviation.

Figure 2A does not cite in the main text.

Please, transfer the websites and the online resources from the main text in the reference list.

Table 1, column 1, Lines 3, 4, 5

Please write “no flood” with the same font of the main text.

Line 107

“This episode was emblematic…”

should be

“This event was emblematic…”

COMMENT: In my opinion “event” is more correct term in this case instead of the term “episode”.

Lines 123 – 125

The authors state “3. Meteorological features/ In this Section, the meteorological conditions together with the measured precipitations in the  Salento peninsula are described in relation to the four floods and other major precipitation events,  (Table 1)”.

COMMENT: In fact, this sentence is a repetition of the Introduction (Lines 51-53: "In Section 3, the meteorological conditions in the Salento peninsula in that period are illustrated, with reference to the  precipitations triggering the four floods"). Please, delete the Lines 123-125 and transform the main text suitably.

Line 227

“As reported above, the…” should be “As aforementioned, the..”

Line 228

“….to the Vora Colucce hole (2B). Since …” COMMENT: What does it mean (2B)? Please, clarify.

Line 395

“…. Current Science ….” should be "…. Curr Sci ….".

Lines 398, 420, 459

“… Journal of Hydrology …” should be “…J Hydrol …”

Line 402

“…. Journal of Environmental Planning and Management …” should be “... J Environ Plann Man… “

Line 404, 408 & 409

“…Natural Hazards and Earth System Sciences…” should be  “…Nat. Hazards Earth Syst. Sci…”

Line 407

“….Theoretical and Applied Climatology 2013, 421, 1–20” should be “Theor Appl Climatol 2013, 111, 1–20”

Lines 416, 418

“…Environmental Earth Sciences…” should be “…Environ Earth Sci …:

Lines 425, 427

“… Urban Water Journal…” should be “… Urban Water J…”

Line 429

“… Meteorological Applications …” should be  “… Meteorol Appl …”

Line 431

“…Weather And Forecasting…” should be “…Weather Forecast…”

Lines 442 & 445

“… Quarterly Journal of the Royal Meteorological Society…” should be “… Q J R Meteorol Soc…”

Line 464

“….Journal of Contaminant Hydrology…” should be “….J Contam Hydrol   “

Line 466

“…Environmental Science & Technology…” should be “….Environ Sci Technol…”

Line 474

“….Physics and Chemistry of the Earth, Parts A/B/C…” should be “…Phys Chem Earth, Parts A/B/C…”

Lines 478 & 479 & 496

“….Science of The Total Environment…” should be “…. Sci. Total Environ ….”

Line 481, 482

“….Natural Hazards…” should be “   Nat Hazards …”

Line 492

“… Environmental Science: Water Research and Technology…” should be “…Environ Sci Water Res Technol…”

Author Response

Dear Reviewer,

we express our gratitude for the excellent revision work you made. We have modified the manuscript in accordance with your comments.

The answers to your comments are in what follows.

“The manuscript examines the results of an investigation about the meteorological and hydrogeological causes generating the hydraulic disorder in the Asso Torrent basin, Southern Italy. It is worth to note that the Asso Torrent Basin is a karst region. Consequently, the manuscript examines a topic of interest to several regions worldwide with similar characteristics.

In my opinion, the authors have made an important and serious effort to improve the quality of the manuscript in relation to the previous version (manuscript ID:water-684474). I consider they succeeded it.

However, the manuscript is needed minor revision to be suitable for publication. Consequently, I recommend acceptance of the manuscript after minor revision.

Specifically:

The manuscript has to be written according to the instructions for Authors of the WATER Journal, namely in the reference list the Journal’s name in abbreviation.”

The names of the journals in the reference list have been edited according to the instructions for Authors, also following each specific comments (see below). Many thanks.

“Figure 2A does not cite in the main text.”

The citation are now at the line 65.

“Please, transfer the websites and the online resources from the main text in the reference list.”

We have transfer they in References.

“Table 1, column 1, Lines 3, 4, 5

Please write “no flood” with the same font of the main text.”

We have edited “no flood” using the same font of the main text.

“Line 107

“This episode was emblematic…”

should be

“This event was emblematic…”

COMMENT: In my opinion “event” is more correct term in this case instead of the term “episode”.

We agree with your comment and we have made the replacement of the term.

“Lines 123 – 125

The authors state “3. Meteorological features/ In this Section, the meteorological conditions together with the measured precipitations in the  Salento peninsula are described in relation to the four floods and other major precipitation events,  (Table 1)”.

COMMENT: In fact, this sentence is a repetition of the Introduction (Lines 51-53: "In Section 3, the meteorological conditions in the Salento peninsula in that period are illustrated, with reference to the  precipitations triggering the four floods"). Please, delete the Lines 123-125 and transform the main text suitably.”

We apologize for the repetition. We have transformed both the phrases.

“Line 227

“As reported above, the…” should be “As aforementioned, the..””

This change has been made.

“Line 228

“….to the Vora Colucce hole (2B). Since …” COMMENT: What does it mean (2B)? Please, clarify.”

It was an omission, specifically of the word "Figure". Correction has been made, please see line 220 of the final version.

“Line 395

“…. Current Science ….” should be "…. Curr Sci ….".

Lines 398, 420, 459

“… Journal of Hydrology …” should be “…J Hydrol …”

Line 402

“…. Journal of Environmental Planning and Management …” should be “... J Environ Plann Man… “

Line 404, 408 & 409

“…Natural Hazards and Earth System Sciences…” should be  “…Nat. Hazards Earth Syst. Sci…”

Line 407

“….Theoretical and Applied Climatology 2013, 421, 1–20” should be “Theor Appl Climatol 2013111, 1–20”

Lines 416, 418

“…Environmental Earth Sciences…” should be “…Environ Earth Sci …:

Lines 425, 427

“… Urban Water Journal…” should be “… Urban Water J…”

Line 429

“… Meteorological Applications …” should be  “… Meteorol Appl …”

Line 431

“…Weather And Forecasting…” should be “…Weather Forecast…”

Lines 442 & 445

“… Quarterly Journal of the Royal Meteorological Society…” should be “… Q J R Meteorol Soc…”

Line 464

“….Journal of Contaminant Hydrology…” should be “….J Contam Hydrol   “

Line 466

“…Environmental Science & Technology…” should be “….Environ Sci Technol…”

Line 474

“….Physics and Chemistry of the Earth, Parts A/B/C…” should be “…Phys Chem Earth, Parts A/B/C…”

Lines 478 & 479 & 496

“….Science of The Total Environment…” should be “…. Sci. Total Environ ….”

Line 481, 482

“….Natural Hazards…” should be “   Nat Hazards …”

Line 492

“… Environmental Science: Water Research and Technology…” should be “…Environ Sci Water Res Technol…””

The names of the journals in the reference list have been edited according to the instructions for Authors, also following each specific comments.

We are grateful for your work. Your comments helped us to improve the manuscript. We hope that it is now suitable for publication in Water.

Reviewer 2 Report

The Authors have done much efforts trying to solve questions raised during the first round of reviews. The paper is more readable and improvements are evident. Topic and targets of the manuscript are much more clearer than before. Only few points should be modified before acceptance:

1) Fig.2 should contain hydrogeology (i.e. hydrogeological complexes) and a hydrogeological cross-section;

2) Fig. 6 should be improved.

Author Response

Dear Reviewer,

we express our gratitude for the revision work you made. We have modified the manuscript in accordance with your comments.

The answers to your comments are in what follows.

“The Authors have done much efforts trying to solve questions raised during the first round of reviews. The paper is more readable and improvements are evident. Topic and targets of the manuscript are much more clearer than before. Only few points should be modified before acceptance:

1) Fig.2 should contain hydrogeology (i.e. hydrogeological complexes) and a hydrogeological cross-section;”

We agree with this comment. However, we have described the hydrogeological setting of the basin in our previous works (listed in References as 10 and 11). Thus, we have provide this appropriate reference by citations at the line 59 of the edited version.

“2) Fig. 6 should be improved.”

Specially thanks for this comment. We have modified Figure 6 as you see at page 9. We believe that it has now graphically improved.

We are grateful for your work. Your comments helped us to improve the manuscript. We hope that it is now suitable for publication in Water.

This manuscript is a resubmission of an earlier submission. The following is a list of the peer review reports and author responses from that submission.

Round 1

Reviewer 1 Report

The manuscript entitled “The recent floods in the Asso Torrent basin (Apulia, Italy): an investigation to improve the stormwater management”  by Delle Rose and co-Authors deals on the recent floods which have taken place on a small river (named Asso) located in southern Italy. Floods occurred in a time-window that lasted from November 2013 and October 2018; they were triggered by rainfall events characterized by different intensities and soil moistures (which varied from 25% to 80%, as registered from 1 monitoring point). Differently from the majority of study-cases, there the outlet of the river is represented by several ponds connected to karst holes feeding an important aquifer that currently provides water also for human purposes. As the Authors stated at the beginning,  due to both climate change and increase of population ( I feel to suggest also a wrong landuse planning), the river basin is characterized by high hydraulic disorder that in turn leads to a remarkable hydraulic risk (even considering a low return period). Moreover, pollutants  may affect the aquifer hosted in this karst formation.

The main objective of the work is thus to “investigate the meteorological and hydrogeological causes generating the hydraulic disorder in the Asso Torrent basin are reported. They might be considered when defining new management solutions and alternative criteria of the alert system“.

I anticipate that this case study (coupled with the abovementioned target of the work) would be of  interest for the Journal readership.

Anyway, I do not feel to be positive with this work, which, at the end,  left me bitter disappointed as, in a very poor way, qualitatively assess the main causes of the flooding in 1) pre-event soil saturation, 2) cumulative precipitation 3) clogging of the shallow holes.  By looking carefully within the work, I found  deeply treated the only weather conditions that have led to the rainfall events; this is probably because the no-other fundamental data (those related to hydrology-hydrogeology) were available (only 1 hydrometric station, 1 soil moisture probe and few meteorological stations).

I suggest to improve the work by enlarging the datasets of historic floods (true alarm)/no floods (flase alarm) dataset and the corresponding weather conditions that have led the floods/no floods.

If the Authors want to define an alert system focused on the meteorological condition (I.e. rainfall thresholds) affecting the basin, they should start  from the weather stations located in the area and carefully compare some parameters with the local flooding history: Meteorological-related parameters such as, for example, and without claims to be exhaustive:

average intensity antecedent moisture conditions (API index?) peak intensities variations total rainfall

may be calculated and compared with floods in order to provide rainfall thresholds (the only alert system that could be implemented in case of poorly-monitored basins).

A number of works have dealt with flood thresholds assessment, even with ungauged basins. Here, I would recall the following:

Diakakis, M. (2012). Rainfall thresholds for flood triggering. The case of Marathonas in Greece. Natural hazards, 60(3), 789-800.

Montesarchio, V., Napolitano, F., Rianna, M., Ridolfi, E., Russo, F., & Sebastianelli, S. (2015). Comparison of methodologies for flood rainfall thresholds estimation. Natural Hazards, 75(1), 909-934.

Reviewer 2 Report

General comments

Τhe manuscript examines the results of an investigation about the meteorological and hydrogeological causes generating the hydraulic disorder in the Asso Torrent basin, Southern Italy.

However, the Asso Torrent Basin is a karst region in which several studies have taken place concerning its karstic character, and the geomorphological and hydrological features. The manuscript does not provide any background of these studies and it does not mention similar studies in the area.

Τhe introduction should be expanded and several literature references will need to be added. Some examples of similar research works could be mentioned either, in Apulia, Southern Italy or internationally for similar cases.

Italy has an official national plan aimed at reducing the risks of floods  in the most vulnerable parts of its territory in accordance the floods directive (Directive 2007/60/EC). Apulia is such area, vulnerable in floods. Therefore, I think the article will become more attractive to the international reader, if information is added to the predicted measures in the area of this national plan.

An area of about 16500 km2 (percentage 5.5 %) of the Italian territory is currently classified at risk of desertification, mostly localized in the South of the country, namely Apulia, Basilicata, Calabria, Sicily, and Sardinia. I consider the authors have to take into account in their study this situation and to comment it.

In reality, the manuscript describes 4 events occurred between 2013 and 2018 and it gives general information about the region and of its characteristics. I cannot see the novelty of the manuscript. Generally speaking, it is about a report and not a research article.

Moreover, the manuscript has to be written according to the instructions for Authors of the WATER Journal. Also, all Figures, Schemes and Tables should be inserted into the main text close to their first citation and must be numbered following their number of appearance (Figure 1, Scheme I, Figure 2, Scheme II, Table 1, etc.). In the manuscript, all the Figures and Tables are inserted into the main text before their first citation.

Specific comments

Lines 9 - 11

The authors state: “With a view to providing solutions for a sustainable management of the water resources, new measures are advanced, such as innovative agricultural practices for flood-tolerant crops and construction of harvesting systems for alternative water uses”. COMMENT: By whom are they promoted these measures? Please, clarify.

Line 12

I would recommend the authors as keywords.

“water resources management; floods; swallow holes; endorheic basin; karst; Apulia (Southern Italy)”

instead of

“sustainable water management; floods; hydraulic disorder; swallow holes”

Lines 20 & 21

The authors state: “thus in 2013 more consistent works were executed, essentially for the protection of the urban areas”.

COMMENT: Give examples of what "consistent works" you are referring to.

Table 1

Please, clarify the terms “low degree of damage” and “middle degree of damage”. What do you mean?

Lines 56 & 57

“The discharge capacity of the swallow hole reduced from 10 m3/s to 1m3/s in thirty years (Please, add citation)”.

Lines 101 & 102

The authors state: “…other major precipitation events, all occurred between 2013 and 2018..”

COMMENT: While the literature shows that there have been more significant rainfall events in the area in the past, the authors do not take them into account. Why do they limit their interest in the period 2013- 2018.

As example the following references

Parise, M. Flood history in the karst environment of Castellana-Grotte (Apulia, southern Italy). Natural Hazards and Earth System Sciences 2003, 3, 1–12. Polemio, M. Historical floods and a recent extreme rainfall event in the Murgia karstic environment (Southern Italy). Zeitschrift fÏŒr Geomorphologie 2010, 54(2), 195-219 Delle Rose, M.; Parise, M. Water management in the karst of Apulia, southern Italy. In Sustainability of the karst environment, Proceedings of International Interdisciplinary Scientific Conference, Plitvice Lakes, Croatia, September 23-26, 2009; O. Bonacci, Ed.; UNESCO, IHP-VII/2010/GW-2: Paris, France, 2010, pp. 33-40.

Line 110; Figure 4, Caption; Figure 6, Caption

The websites have to be transferred in the reference list and to substitute by numbers in brackets in the main text.

Lines 185 & 186

The authors state: “The other relevant events (not associated to a flood g) in terms of enhanced rain rate in few days  registered by the ISAC-CNR station in the five-years period 2015-2018 are the following”.

COMMENT: I do not understand why this information added. Which is its role in the total problem.  I think it has nothing to do with the subject. Please, clarify the reasons.

Lines 205 & 206

“(located just upstream the gate G2, see Section  2, Figure 2B, Figure 4A)”

should be

“(located just upstream the gate G2, see Figures 2B and 4A)”

Line 214

“Contrary to this case…” should be “In contrast to this case…”

Line 344

“…doi:https://doi.org/10.1016/j.scitotenv.2009.01.022.”

should be

“…https://doi.org/10.1016/j.scitotenv.2009.01.022.”

COMMENT: Please, delete “doi:”

Line 314

“….doi:doi:10.1080/09640568.2019.1634015…” should be “….doi:10.1080/09640568.2019.1634015…”

COMMENT: Please, delete “doi:”